# OpenReview forum: "Uncertainty-Driven Exploration for Generalization in Reinforcement Learning"
_ICLR.cc/2023/Conference — Submitted to ICLR 2023_

### Official Review · Reviewer_WM6p · 2022-10-19

**Confidence:** 3
**Correctness:** 3
**Technical Novelty And Significance:** 2
**Empirical Novelty And Significance:** 3
**Recommendation:** 5

**Clarity, Quality, Novelty And Reproducibility:**

There are technical issues, inaccuracies, or misleading sentences throughout. Most of these are minor enough that they could be easily fixed, but some require more attention. A lot of terms are used without being explicitly explained or defined, in both the Figures and the main text.

Specific points:

**Introduction + Background**:

- It would be much better to provide an explanation, even a minimal one, for *what* are Value-based and Policy-optimization methods and how do they differ **in the introduction**. The terms are used freely in the Introduction and then only explained in the "Background" section. It would be enough to include one or two citations to some classic works on value-based / PG (or perhaps a review, e.g. , instead of the long lists of more recent cited works.

- The explanation of PG methods is lacking and would be totally unhelpful (and even misleading) for someone who's not already well familiar with them. Most crucially, it should be mentioned that there is a policy *parameterization*, what is the involved "gradient", add some more specific references other than the generic textbook one. e.g., Williams 92 or Sutton 99)

- The indicator function notation for a greedy policy (which is used throughout) is a confusing abuse of notation (for example, the same $a$ is used on LHS and in the argmax so the RHS is the same regardless of the action). The argument inside the indicator is not even a logical condition but just an action. It is better to simply explain in words, or use a better notation (defined by cases, for example).

**Tabular**:
-  The "suboptimality" measure shown in Figure 2 is only defined somewhere in the appendix. Should be clearly defined in the main text.

- it is not clear how the policy is parameterized for the PG agent (direct representations of the probabilities, with a projection step? Softmax?) It is only mentioned in the appendix that it's softmax but should defenitely be included in the main text.

- "greedy is naive and only works for simple environments":  the second part is misleading. The original DQN works were entirely based on $\epsilon$-greedy exploration, and performed well on many non-trivial environments, despite the exploration strategy being, indeed, naive.

- The sentence about Boltzman exploration (another term which is used but not defined; in addition other instances in the paper uses "Softmax" and not "Boltzman") is misplaced or out of place.

- **Important**: the use of a "plug-and-play" version of the standard bandit UCB here is wrong, and is inconsistent with the rest of the paper. It is definitely not the method in the cited Chen et al. 2017 (which uses ensembles rather than straight visit counters). This use of counters might be by itself  a rather naive form of exploration (e.g. Fox et al. 2018)

**Exploration via distributional ensemble**
- What is the justification for using a Normal posterior in the Thompson sampling action-selection? is this an approximation? Just few sentences before it is mentioned that "the unimodal nature of a Gaussian in parameter space may not be sufficient for effective uncertainty estimation", so why here it is used again?
- In general this section, particularly "Uncertainty estimation" is hard to follow, and does not feel enough self-contained. This describes the key elements in the proposed architecture of the paper and should be more accessible.

**Experiments**
- The measures used for evaluation should be explained explicitly, especially as these are not (yet) very standard.
- Related, the use of both "optimiality gap" and "mean" is redundant here as optimality gap = 1 - mean (assuming the $\gamma$ threshold is 1; see Agarwal et al. 2021 Fig. A.32 ).
- The "sample efficiency" paragraph: It is hard to draw any conclusions if we don't see the comparison of DQN and PPO on the "contextualized" settings. There's no PPO in Fig. 4, maybe it's about as good as DQN there too?
- The "partial observability" paragraph: the citation of Singh 94 is misleading, as it has nothing to do with policy optimization vs. value-based methods (and the policy of value-based methods can also be stochastic, depending on the action-selection rule). Partial observability surely affect PG methods as well. Another term ("dynamic programming") is used here out of context.
	- The task-ID experiment is rather strange. First, it is entirely not obvious that "each individual environment $\mu$ is Markovian" on the level of visual inputs (This is not even true for Atari games in general) and when a complicated function-approximator is used. The different environments are not different "tasks", so the taskID information is indeed kind of useless here for the network, I find it hard to understand why the paper mentions that it is "surprising" that this doesn't help training. In general, the conclusion or claim that "partial observability may not be the main problem in such environments" is another strong claim that is far from being sufficiently supported.

**Related work**
- The paper does not deal at all with "exploration-exploitation tradeoff" so it's not clear why this is the way the paragraph about "Exploration" starts. Rather than providing a long list of tangentially related exploration papers (which can never be exhaustive, and since this is not a review on exploration, nor should it be) it is better to spend the place in explaining the how the paper actually *relates* to the literature (it is currently somewhat there, but gets lost among many non-relevant references)

- "Our work is the first one to highlight the role of exploration for faster training on contextual MDPs and better generalization to unseen MDPs" -- this is certainly inaccurate. Many works on "task agnostic RL", or "multitask RL" and so on have emphasized the importance of exploration for task adaptation, etc. See for example Zhang 2020, Lee 2020 (and references therein), the ICLR workshop on task-agnostic RL, etc.

**Strength And Weaknesses:**

The paper tackles a challenging benchmark and has good evaluation methodology, including an ablation study for the different components of the architecture, and comparison with strong baselines. The empirical results are strong, or at least competitive with the state-of-the-art on an aggregate level.

On the other hand, the main conceptual message of the paper seems confused, and some claims are not well supported by neither theory nor empirical evaluation (see main issues below). Moreover the large heterogeneity of performance on individual task basis (Appendix D) makes the empirical results much less conclusive.

Main issues:

* The distinction between "Value-based" and  "policy optimization" (I will refer to these as PG, for policy gradient) which is emphasized repeatedly in the paper seems a bit arbitrary. First, most (if not all) PG methods discussed and compared to are some kind of an actor-critic ("hybrid") architecture which does, in fact, involve Value estimation/learning.

* The tabular example does not seem to support the later claims at all.
	* First, the claim that (in Procgen) "the agent can often end up in states that  are suboptimal for the training MDPs." is far from being trivial and it requires at least a convincing demonstration by examples. It seems that the main issue in Procgen is generalization in terms of good "semantic" feature extraction for the state structure (identifying platforms from background, enemies, etc.) rather than learning to adapt to a "new task" in the same environment (e.g., different target or initial state).
	* The exploration of the PG agent is by no means more sophisticated than that of the $\epsilon$-greedy agent, both rely on purely random (and uniform) exploration (moreover, the entropy regularization have no real effect here, as is seen clearly in Fig. 6c). The main reason limiting e-greedy compared to PG here is most likely the reliance on 1-step updates versus trajectory-based updates. In PG, after each rewarding episode the policy for states along the entire trajectory will be updated. By contrast, the simple Q-learning has to "propagate" values back from the rewarding state. But, and related to the previous point, this is not an essential limitation of *value*-based (for example, TD($\lambda$) will not suffer from this).

* Some strong claims are made about exploration and its components but the supporting evidence is too weak. Importantly, this kind of claims are very hard to support based on some aggerated performance measure alone, as they are really claims about the behavior and learning of the agents (in specific tasks). Most conclusions about why Alg1 > Alg2 are not much more than interpretations, which may or may not be a key reason.
  Particular examples:
	- The mentioned "importance of Distributional RL" (which seems not inline with NoisyNet performance, for example)
	- The ez-greedy "not accounting for what the model already knows" as opposed to the proposed architecture.
	- Even the fact that the learning is successful in reducing the epistemic uncertainty is not exemplified.
 More generally speaking, the fact that "exploration as a key factor for good sample efficiency and generalization" is almost a triviality. And surely it doesn't provide an answer for the question "why value-based approaches work well in singleton MDPs but not in contextual MDPs". (see previous points as well).

**Summary Of The Paper:**

The paper proposes a value-based RL architecture that is aimed at achieving good generalization, in the settings of "contextualized" MDPs (where the agent is trained on one set of MDPs and tested on a different set sampled from the same distribution). This is achieved by encouraging a more explorative behavior during training. The architecture itself builds on and combines concepts from distributional RL, ensemble training, and classic exploration approaches (UCB-like and Thompson sampling). The method is evaluated on two benchmarks of procedurally generated RL environments.

**Summary Of The Review:**

The architecture proposed in the paper seems effective and the results are overall promising. However at its current form the paper is missing a coherent (and supported) main message. It offers some general high-level intuitions about the importance of exploration, but how and if these are actually related to the proposed performance of the algorithm is unclear, and probably cannot be deduced based on performance metrics alone (without considering the behavior, and/or some more direct measure for its "explorativitiy"). Taken together with the technical issues, the paper seems not ready for publication at its current stage.


**EDIT**
The revised manuscript is an improvement, and the authors have addressed most of the technical issues raised in my review.
I've changed my recommendation to reflect these improvements.

---

> ### Author Response · Authors · 2022-11-09
> **Response to Reviewer WM6p 1/4**
>
> We thank the reviewer for their detailed and helpful comments which have us improve the paper. We were glad to hear that they consider our paper to “tackle a challenging benchmark”, “have good evaluation methodology”, and “strong empirical results”.
>
> ### Main Concerns
>
> > First, most (if not all) PG methods discussed and compared to are some kind of an actor-critic ("hybrid") architecture which does, in fact, involve Value estimation/learning.
> >
> We completely agree that the PG methods also learn value functions and that the lines can become blurry for certain algorithms. We would like to note that chapter 13 of Sutton & Barto classifies actor-critic as a policy gradient method, which is consistent with our use of this term. In Section 2, we explain that by value-based methods we mean those that directly use the value to define the policy as opposed to having another function that learns a probability distribution over the actions (potentially, in addition to learning a value function). We have also added this to the introduction now.
>
> > the claim that (in Procgen) "the agent can often end up in states that are suboptimal for the training MDPs." is far from being trivial and it requires at least a convincing demonstration by examples...adapt to a "new task" in the same environment
> >
> We agree with the reviewer that this sentence represents our intuition/hypothesis rather than a well-supported fact and we have clarified in the text that it’s a hypothesis. However, it is quite difficult to prove this on Procgen exactly given the high dimensionality of the observations, substantial diversity of the levels, and lack of access to the internal state of the simulator. As an alternative, to provide more intuition about this phenomenon, we added a detailed illustrative example in Appendix I, which is based on Bigfish, one of the Procgen games. In addition, we would like to point out that our hypothesis applies even if the model has already learned a good semantic representation (e.g., low-dimensional tabular representation). We have made this point clearer in Sec3.
>
> > The exploration of the PG agent is by no means more sophisticated than that of the ϵ-greedy agent
> >
> We do not intend to suggest that PG’s exploration is more sophisticated than $\epsilon$-greedy. We made some changes to Section 3 to avoid any confusion around this. The main goal of these experiments is to contrast different exploration methods used with Q-learning and show that a more effective exploration approach improves generalization. Comparing the exploration of PG and value-based methods is not trivial and we do not claim to fully understand how they differ.
>
> >The main reason limiting e-greedy compared to PG here is most likely the reliance on 1-step updates versus trajectory-based updates…this is not an essential limitation of value-based (for example, TD(λ) will not suffer from this)
> >
> This is an astute observation and one that we only realized after the submission. We have added additional tabular experiments on Sarsa$(\lambda)$ in Sec 3 which improves upon both epsilon-greedy and PG but it still underperforms UCB, which suggests exploration may still offer benefit even for trajectory-based update. This is an interesting point that warrants future research but we believe this is orthogonal to our claim that exploration can improve generalization, especially since there is still a gap between Sarsa$(\lambda)$ and UCB.
>
> On the other hand, our experiments are meant to highlight how many existing value-based methods in deep RL behave, which is indeed more like 1-step TD. To the best of our knowledge, TD$(\lambda)$ is not widely used in deep RL for the environments we consider, which is ultimately the goal of our paper. Further, to use TD$(\lambda)$, the algorithms are usually on-policy, whereas Q-learning can be done off-policy. Again, the focus of our paper is not to claim that PG methods generalize better than value-based methods because they better explore their training environments. Rather, _our focus is on improving the generalization of value-based methods by employing a more effective exploration strategy during training._ We apologize for any confusion and we changed the text in Sections 1 and 3 to better reflect this.

---

> > ### Author Response · Authors · 2022-11-09
> > **Response to Reviewer WM6p 2/4**
> >
> > > The mentioned "importance of Distributional RL" (which seems not inline with NoisyNet performance, for example)
> > >
> > We are not sure what the reviewer means here since our results show that distributional RL outperforms NoisyNet (Figure 4). Is this not sufficient evidence that distributional RL is important? Furthermore, the full claim is “_the importance of using distributional RL in CMDPs in order to measure epistemic uncertainty_” which we believe is justified given that the methods that estimate uncertainty without the disentanglement are much worse. We have not claimed that methods cannot succeed without distributional RL which would be of course incorrect.
> >
> > > The ez-greedy "not accounting for what the model already knows" as opposed to the proposed architecture.
> > >
> > We are not sure what the reviewer means here. By construction, ez-greezy does not use epistemic uncertainty to explore the environment, so it does not account for the model’s knowledge during exploration. We have added clarification to explain what we mean. In contrast, EDE accounts for what the model knows by using epistemic uncertainty.
> >
> > > Even the fact that the learning is successful in reducing the epistemic uncertainty is not exemplified.
> > >
> > The idealized goal is to ultimately reduce the epistemic uncertainty but it is not the case in general. While it is true that asymptomatically, if we have collected enough data and have an expressive enough function class, then the epistemic uncertainty would eventually decrease. However, in the finite sample case, the trend of epistemic uncertainty can vary depending on the specific CMDPs. For example, if the agent follows action with high epistemic uncertainty, it is likely that it will end up in a state with equal or even higher uncertainty since its predecessor has not been visited before. This means that, for large CMDPs, the average epistemic uncertainty of observed states can actually _increase_, or _stay the same_ until the end of training compared to $\epsilon$-greedy if we explore with epistemic uncertainty, which we do observe in some CMDPs. Nonetheless, the performance on these CMDPs still improves because the agent is exploring the environment efficiently. Note that there is no direct relationship between low average epistemic uncertainty and good performance, but rather that trying to reduce epistemic uncertainty can lead to good exploration which in turn leads to good performance. We verify this in Figure 12.c where we can see exploring with epistemic uncertainty is much better than exploring with total uncertainty. On the other hand, a bad exploration policy can have low average epistemic uncertainty in its observed state if it only ever sees a small number of states.
> >
> > > exploration as a key factor for good sample efficiency and generalization" is almost a triviality.
> > >
> > While it is broadly accepted that exploration is essential for good sample efficiency, we are not aware of any work that shows that better exploration of the training environments leads to better zero-shot generalization to new environments from the same family, when learning is in CMDPs like the ones we consider in this work. Can the reviewer point us to any relevant prior work? If the concern is about sample efficiency, we have removed sample efficiency from this sentence.

---

> > > ### Author Response · Authors · 2022-11-09
> > > **Response to Reviewer WM6p 3/4**
> > >
> > > > And surely it doesn't provide an answer for the question "why value-based approaches work well in singleton MDPs but not in contextual MDPs"
> > > >
> > > We do not intend to claim to have a definite answer (we have edited the introduction and where applicable to emphasize this), and we suspect there might be other reasons for why value-based methods don’t work as well in CMDPs (such as 1-step update). Indeed, anything short of a proof cannot claim to be a definitive reason. What we do claim is that exploration is one component that affects learning in CMDPs. Our experiments on tabular CMDPs, Procgen, and Crafter, all support this claim by showing that more sophisticated exploration strategy leads to better generalization to new environments since all algorithmic components of EDE are designed to target exploration with uncertainty.
> > >
> > > > this kind of claims are very hard to support based on some aggerated performance measure alone, as they are really claims about the behavior and learning of the agents (in specific tasks).
> > > >
> > > We would like to emphasize that our claim is that exploration can affect generalization and that our proposed exploration method improves generalization performance. We believe both claims are reasonable based on the results (both tabular and deep RL) in the paper. Does the reviewer have more specific suggestions for what experiments to run or what metrics to track in order to demonstrate them? Analyzing the behavior and learning dynamics of agents while training in complex environments such as Procgen or Crafter which have extremely large numbers of effective states is highly non-trivial. For example, it is not clear how to count visited states or measure their diversity in high-dimensional observations. This is why we decided to rely on widely-used, robust, and objective metrics like aggregate performance on two popular generalization benchmarks, as well as a didactic example in a tabular CMDP which is simpler to analyze. This is also how RL methods are evaluated on Atari57.
> > >
> > > Even in individual games (Figure 10), EDE outperforms qr-dqn on 11/16 games and performs about the same on the remaining ones, suggesting that the improvement is robust across different CMDPs and EDE does not hurt the performance even when it doesn’t help. Finally, EDE also significantly helps Crafter which is a very different environment compared to those in Procgen. That being said, we are open to hear your suggestions.
> > >
> > > ### Minor Comments
> > >
> > > #### Introduction + Background
> > > > value-based and policy-optimization
> > > >
> > > Thank you for the suggestion. We modified the introduction to reduce the focus on the  comparison between value-based and policy-optimization and to provide more general citations, as suggested. In Section 2, we define the two terms to clarify their use throughout the paper.
> > >
> > > > Explanation of PG methods
> > > >
> > > Thank you — we added more details in Section 2 as suggested.
> > >
> > > > Indicator function
> > > >
> > > Thank you — we changed the notation to the standard notation of indicator function, $1_A(x)$.
> > >
> > > **Tabular**
> > >
> > > > Suboptimality
> > > >
> > > We define this in the caption but have also added it to the main text.
> > >
> > > > Parameterization for the PG agent
> > > >
> > > We describe this in the appendix but have also added it to the main text.
> > >
> > > > Greedy is naive and only works in simple environments
> > > >
> > > We agree and have modified the claim accordingly.
> > >
> > > > Boltzman exploration vs softmax exploration
> > > >
> > > Thank you for pointing this out — we have changed it to softmax exploration.
> > >
> > > > UCB implementation
> > > >
> > > Thank you for highlighting this — we changed the language to be more accurate and consistent (See footnote in Sec 3). Chen et al. 2017 is for deep RL. In tabular settings, we can use simpler methods for counting the states, which is what we do. We agree this is still a naive approach intended for illustration but our experiments show it is already much better than $\epsilon$-greedy or policy-gradient.
> > >
> > > **Exploration via distributional ensemble**
> > > > Normal posterior in the Thompson sampling
> > > >
> > > Thank you for catching this typo. Indeed, it is an approximation that we can efficiently sample from. The preivous prior we discussed was in the **parameter space** (i.e., the parameters of the networks have a gaussian prior). Here we are using the Gaussian to approximate the posterior in the **function space** (i.e., the quantiles). Diversity in parameter space is needed for expressive posterior approximation but when it comes to sampling, since we already have an estimation of the actual uncertainty via the ensemble, we can use cheaper approximations such as the Gaussian. This was our mistake — we haved changed text to say the prior to be over parameter space instead of quantile. We also added more details about this in Section 4.

---

> > > > ### Author Response · Authors · 2022-11-09
> > > > **Response to Reviewer WM6p 4/4**
> > > >
> > > > > Uncertainty estimation section
> > > > >
> > > > Thank you for noticing this. We edited the section to make it easier to follow and more self-contained.
> > > >
> > > > **Experiments**
> > > > > Evaluation metrics
> > > > >
> > > >  We added more details about these metrics in Section 5.
> > > >
> > > > > Optimality gap and mean
> > > > >
> > > >  yes, you are correct. We simply added it for completeness. We have added an explanation for this in the caption.
> > > >
> > > > > Sample efficiency
> > > > >
> > > > the comparison with PPO is shown in Figure 3. Note that Figure 3 and 4 report results for the same setting, namely test performance on Procgen.
> > > >
> > > > > Partial observability
> > > >
> > > > we moved this paragraph to the appendix since it’s not essential for the main story. By partial observability, we mean that the agent cannot tell which environment it is in. We agree that the individual environments are partially observable like Atari. We modified the text according to your suggestions.
> > > >
> > > > > task-ID experiment
> > > > >
> > > > we changed the wording to reflect your feedback. They are different tasks insofar as they are different MDPs. We agree that the IDs are useless in this case, which is proved by this experiment. However, the hypothesis being verified here is that with the ID the model would at least be able to fit each environment individually like singletons, which turns out to be not the case.
> > > >
> > > > **Related Work**
> > > > > Exploration-exploitation tradeoff
> > > > >
> > > > thank you for the suggestion. We rewrote this section to reflect your feedback.
> > > > > Exploration for task adaptation
> > > > >
> > > > we added more references and a discussion on how our work differs from others that look at the effect of exploration for adaptation to new MDPs such as task-agnostic RL.
> > > >
> > > >
> > > > ### Summary
> > > > We once again thank the reviewer for their detailed feedback which has helped us greatly improve the paper. We hope that the clarifications above and the changes we made are sufficient for you to reconsider your assessment of the paper. If you have any outstanding concerns that prevent you from recommending acceptance, please don’t hesitate to let us know so we can discuss them in due time.

---

> ### Author Response · Authors · 2022-11-15
> **Have our revision and response addressed the concerns?**
>
> Dear Reviewer WM6p,
>
> Thank you again for the suggestions and questions. They have greatly improved the paper. We believe our revision, new tabular experiments, and response have made the paper and its core messages much clearer. Since the discussion period is closing at the end of the week, we were wondering if your concerns have been addressed? If not, we would be happy to continue the discussion and/or revise the paper.

---

### Official Review · Reviewer_iatu · 2022-10-22

**Confidence:** 4
**Correctness:** 3
**Technical Novelty And Significance:** 2
**Empirical Novelty And Significance:** 3
**Recommendation:** 5

**Clarity, Quality, Novelty And Reproducibility:**

* Clarity: Ok but some statements are problematic.
* Quality and Novelty: The quality is good and the proposed method is novel.
* Reproducibility: The authors provide implementation details but do not provide code. They promise to open-source the code during discussion period.

**Strength And Weaknesses:**

**Strength:**

* Solid experiments
* Significant empirical improvements
* EDE would serve as a strong baseline for future works and be of importance to the RL generalization community.

**Weaknesses:**

The connection between "more effective exploration during training" and "better generalization in testing" is weak. I understand Sec.3 is meant to build this connection. However, since the tabular example is quite far from the realistic setting in Procgen, it is hard to say this is the reason why EDE works well in Procgen. In fact, from Tab.2&3, it seems that the improvements of EDE on testing levels are highly correlated to the improvements on training levels. So a more plausible logic flow is: better exploration -> better performance on training levels -> better zero-shot generalization on testing levels.

Despite the weaknesses, I do think the empirical contributions of this paper are good on their own. So I would suggest the authors reposition the paper from a different perspective.

**Minor issues:**

* In ablation results in Sec.5.1, please add reference to Fig.4.
* Some statements are not grounded and may mislead readers.
  * First paragraph in introduction section: "In this work, we aim to understand why value-based approaches work well in singleton MDPs but not in contextual MDPs". The statement that value-based approaches does not work well in contextual MDPs is unsupported. The only seeming evidence in the context is that most existing SOTA methods on Procgen are based on PPO. But this is probably because the baseline code released from Procgen authors is PPO.
  * Second paragraph in introduction section: "the main goal of exploration is to learn an optimal policy for that environment". This is not correct. Exploration is also useful in the absence of reward. From the context, I assume the authors want to make a contrast that exploration in singleton env can only benefits that environment while in CMDP exploration in one env may also help other envs. However, I see this more of the difference between singleton env and multiple envs. In CMDP, anything we change in one env will likely affect the behaviour in another env. In short, this is not specific to exploration.

**Summary Of The Paper:**

This paper proposes a value-based RL method (termed EDE) to improve generalization in procedurally generated environments such as Procgen. In particular, the authors introduce two novel modifications to QR-DQN: UCB exploration with learned epistemic uncertainty (UCB) and temporally equalized exploration (TEE), which greatly boosts the zero-shot test performance on Procgen.

**Summary Of The Review:**

In summary, I think the paper makes good contributions to the field, but requires some modifications before acceptance.

---

> ### Author Response · Authors · 2022-11-09
> **Response to Reviewer iatu 1/2**
>
> We thank the reviewer for their careful reading of our paper and helpful suggestions. We were glad to hear that they found our method to be “novel”, and our work to be “of importance to the RL generalization community”, with “solid experiments” and “significant empirical improvements”.
>
> ### Main Concerns
>
> >since the tabular example is quite far from the realistic setting in Procgen, it is hard to say this is the reason why EDE works well in Procgen
> >
> The tabular experiments are meant as a motivational example and we do not claim they capture every aspect of more complex benchmarks such as Procgen. Indeed, most tabular environments cannot capture everything about deep RL. Our goal is to highlight that the problem of generalization in deep RL goes beyond representation learning (as typically treated in prior work). That being said, we have included a new section in Appendix I that illustrates how scenarios in the tabular setting can transfer to Procgen using the game bigfish as a concrete example. We hope that this material adds additional support for how the tabular example may be relevant for Procgen.
>
> Note that all the algorithmic details for EDE are specifically designed for estimating uncertainty estimation and using it for exploration, and there are no additional tricks such as data augmentation. On Procgen we show that EDE outperforms methods with less sophisticated exploration strategies during both training and testing (Figure 4). We believe these results, at the minimum, support the claim that better exploration can indeed improve both sample efficiency and generalization, even on challenging deep RL benchmarks.
>
> >the improvements of EDE on testing levels are highly correlated to the improvements on training levels. So a more plausible logic flow is: better exploration -> better performance on training levels -> better zero-shot generalization on testing levels
> >
> This is indeed a valid hypothesis, and we actually already discussed this phenomenon in Section 5.3 which has now been moved to Appendix E. We observed that this phenomenon is environment-dependent. We have a shorter but more explicit discussion of this possibility at the end of Sec 3. We have also added this mention of this in the introduction.
>
> In short, this hypothesis implicitly assumes that the neural network will always learn a good representation for generalization which may or may not be true. With this oracular representation, the agent can learn how to act in unseen situations that have the same representation as the ones seen during training. But what happens when there is a genuinely new situation at test time? This is the situation our hypothesis aims to address. Exploration can also help deal with these genuinely new situations by explicitly inducing them during training. Tabular environments in some sense represent these “oracular” representations. As our tabular experiments (Section 3) show, even when having such “oracular” representations, better exploration still improves test performance on a new MDP, suggesting that exploration may directly affect generalization. More importantly, these two hypotheses can both be **simultaneously true** and we have adjusted the text to reflect this (Introduction and Sec 3).
>
> Nevertheless, we believe that the claim that “more effective exploration during training leads to better generalization” holds since our results indicate that more sophisticated exploration methods can lead to higher train and test performance, which implies better zero-shot generalization to new environments. Furthermore, previous exploration methods do not do well in these situations, corroborating our hypothesis that exploration in CMDPs has additional challenges not present in singleton MDPs.
>
> ### Minor Comments
>
> > The authors provide implementation details but do not provide code.
> >
> Apologies for the delay, we wanted to clean up the code before uploading it. We have now added the code in the supplementary material, along with instructions for reproducing our results.
>
> > The statement that value-based approaches does not work well in contextual MDPs is unsupported.
> >
> Thank you for pointing this out. We agree with the reviewer that we did not provide enough evidence for this claim in the paper. However, we believe this sentence is well supported by [1] and [2], both of which show that value-based methods like QR-DQN and Rainbow underperform policy optimization ones like PPO on CMDPs such as Procgen. We modified our claim to make it more precise and added these references to support it.

---

> > ### Author Response · Authors · 2022-11-09
> > **Response to Reviewer iatu 2/2**
> >
> > >Exploration is also useful in the absence of reward.
> > >
> > Thank you for highlighting this. We agree with the reviewer that exploration is also useful in the absence of reward. However, in this work, we are concerned with training on environments with external rewards, hence why our claim is specific to this case. Nevertheless, we modified the claim to be more broadly accurate and focused on the distinction between training on single vs multiple environments (with or without external rewards) and added this to related works.
> >
> > > add reference to Fig
> > >
> > Thank you for the suggestion — we have added the reference.
> >
> > ### Summary
> > We thank the reviewer again for valuable questions and comments that have further improved the quality of our paper. We hope that the clarifications above and changes to our paper are sufficient for you to consider increasing your support for our paper. Please let us know in case you have any further questions or concerns that stand between us and your strong recommendation for acceptance.
> >
> > ### References
> > [1] Improving Generalization in Reinforcement Learning with Mixture Regularization. Wang et al., NeurIPS 2020.
> >
> > [2] A Study of Off-Policy Learning in Environments with Procedural Content Generation. Ehrenberg et al., ALOE Workshop at ICLR 2022.

---

> ### Author Response · Authors · 2022-11-15
> **Have the revision and response addressed the concerns?**
>
> Dear Reviewer iatu,
>
> Thank you for the suggestions for improving the paper. Our revision has put more emphasis on the “better training performance leads to better generalization” hypothesis and addressed other concerns you brought up. Together with the discussion above, have all the concerns been addressed? If not, we would be happy to continue the discussion and/or revise the paper.

---

> ### Author Response · Authors · 2022-12-09
> **Follow-up**
>
> Since the discussion is closing soon, we were wondering if our revision has addressed your concerns. To summarize, we have revised the text to discuss the different hypotheses of how exploration improves generalization in more detail (both were already in the original text). In short, both hypotheses can be simultaneously true, and proper exploration can address both, but they consider **two different kinds of generalization**. Please see our response below (and the text) for more details. If you have any remaining concerns, we would be happy to continue the discussion. On the other hand, if our revision of the paper is satisfactory, would you consider improving your support for our paper?

---

### Official Review · Reviewer_R7TS · 2022-10-23

**Confidence:** 4
**Correctness:** 3
**Technical Novelty And Significance:** 2
**Empirical Novelty And Significance:** 3
**Recommendation:** 6

**Clarity, Quality, Novelty And Reproducibility:**

The paper is well-written in general (question #3 mentioned a part that is unclear to me) and the experiments are convincing. The novelty of this paper is quite limited since the uncertainty estimation techniques are adapted from prior work, and the primary contribution of the paper is to successfully apply such techniques to the task of generalization in RL.

**Strength And Weaknesses:**

Strength:

- Although the idea of separating aleatoric and epistemic uncertainty is not new, this paper shows that this idea can be used to improve generalization in RL environments.

- The experiments conducted in the paper are comprehensive: the diversity of the adopted tasks and baselines is large enough to demonstrate the superiority of the proposed method.

Weaknesses and questions:

- It is unclear to me how the bootstrap learning procedure of the Q function affects the estimation of epistemic uncertainty. The uncertainty estimation process adopted in the paper is well-studied on tasks such as classification and environment model estimation. However, it is unclear to me whether the estimation is still accurate when Q-learning style updates are applied. Therefore, I believe this paper should provide at least some intuition into this.

- Prior work such as SAC and TD3 maintain two value heads to mitigate the value overestimation problem of Q-learning. Since EDE also uses multiple value heads, is it possible that it also implicitly mitigates the value overestimation problem? It would be nice to do an additional ablation study to verify this.

- In the abstract and the introduction, the authors mentioned that there is a gap between policy-based methods and value-based methods. However, the paper does not discuss why policy-based methods work better than value-based methods. And the paper is mainly about the importance of exploration + uncertainty estimation, which can also be applied to policy-based methods. It would be nice to either adjust the story or provide more discussion on why the proposed technique works for value-based methods in general.

**Summary Of The Paper:**

This paper studies the task of generalization in RL tasks defined by contextual MDPs (CMDPs). In contrast to meta-RL tasks, this paper focuses on tasks where the agent should learn a policy that performs well on all MDPs instantiated by a CMDP. The paper argues that exploration is key to improving the performance of RL algorithms in such tasks. Here the goal of exploration is not to find a more rewarding policy for a single task, but to better explore the state space to perform well on test tasks. The proposed EDE algorithm estimates the epistemic uncertainty of the Q function to guide exploration. EDE is evaluated on the Procgen and the Crafter benchmarks.

**Summary Of The Review:**

Although the adopted uncertainty estimation techniques are not new, the paper demonstrates that epistemic uncertainty is useful to guide exploration to improve generalization in RL. The effect of uncertainty estimation under a bootstrapped target (Q function learned by Q-learning) is not discussed in the paper, which in my opinion, is a weakness of the paper.

---

> ### Author Response · Authors · 2022-11-09
> **Response to Reviewer R7TS**
>
> We thank the reviewer for their helpful comments and positive feedback. We were delighted to hear that they found “the paper to be well-written”, “the experiments to be comprehensive” and “the diversity of the tasks and baselines enough to demonstrate the superiority of the proposed method.”
>
> ### Questions
>
> > it is unclear to me whether the estimation is still accurate when Q-learning style updates are applied
> >
> Indeed, the reviewer is right in that when Q-functions are involved the estimation of uncertainty is likely not perfectly accurate. However, the estimates don’t need to be fully accurate for them to be helpful in practice if they point roughly in the right direction, as shown by our empirical results (e.g., the difference between different sizes of ensemble is small). The topic of the quality of uncertainty estimation in RL is involved both theoretically and practically. We have added discussion of this issue in “uncertainty estimation” in Section 4.
>
> > Since EDE also uses multiple value heads, is it possible that it also implicitly mitigates the value overestimation problem?
> >
> Prior works on ensembles in RL show that indeed they can alleviate the problem of value overestimation [1, 2]. Since EDE builds on deep ensembles, we expect that it also mitigates this problem.
>
> > The paper does not discuss why policy-based methods work better than value-based methods
> >
> Thank you for pointing this out. Indeed there is no strong reason to believe that softmax exploration is significantly better than $\epsilon$-greedy. We believe there are likely other reasons that make PG more effective than value-based methods in this setting and we don’t pretend to have all the answers to this question. We have updated the paper to focus on the limitations of value-based methods in CMDPs and how exploration can improve their generalization while avoiding the contrast with PG methods to avoid any confusion regarding our core claims. That being said, we have added a new set of experiments in Sec3 involving Sarsa$(\lambda)$ which demonstrate that the improvement of PG might be coming from the whole-trajectory update as opposed to 1-step TD. Future research could expand on this observation.
>
> > The effect of uncertainty estimation under a bootstrapped target (Q function learned by Q-learning) is not discussed in the paper, which in my opinion, is a weakness of the paper.
> >
> Thank you for the suggestion. We added more discussion and relevant literature about this in Section 4 under uncertainty estimation. We would also like to add that this topic is very complicated due to the various components of deep RL, and the community’s understanding of it is quite rudimentary.
>
> ### Novelty
> > The novelty of this paper is quite limited
> >
> We respectfully disagree with this assessment. First of all, the insight that exploration improves generalization on CMDPs is novel. As the reviewer also notes, our paper is the first to show that more effective exploration methods improve generalization to new environments on two different benchmarks. While individual parts of our proposed approach have been used in prior work, this particular combination is novel. In section, 5.1, we run an extensive ablation study and demonstrate the importance of each design choice and component of our algorithm. As shown in Figure 4, naively applying or combining prior exploration methods is not sufficient to significantly improve generalization.
>
> ### Summary
> We thank the reviewer for their helpful questions and comments that led to further improvements to our paper. We hope that we have suitably addressed your concerns with our response and paper edits. Please let us know if there is anything preventing you from improving your support for our paper further so that we can discuss any remaining questions.
>
>  ### Reference
> [1] Reducing Variance in Temporal-Difference Value Estimation via Ensemble of Deep Networks. Liang et al. ICML 22.
>
> [2] Sample efficient deep reinforcement learning via uncertainty estimation. Mai et al. ICLR 22.

---

> ### Author Response · Authors · 2022-12-09
> **Follow-up**
>
> Dear Reviewer R7TS,
>
> Since the discussion period is closing soon, we were wondering if our revision (in particular, the added discussion of the effect of various aspects of deep RL on uncertainty estimation) and response have suitably addressed all of your concerns. Is there anything preventing you from improving your support for our paper further? If so, we would be more than happy to continue the discussion.
>
> We would like to emphasize that the main contribution and novelty of this work is the conceptual connection between exploration and generalization in CMDPs, and how decomposing different uncertainties can achieve the desired exploration. Further, our method significantly outperforms the previous work that uses similar decomposition, so we believe our technical innovation is also non-trivial.

---

### Official Review · Reviewer_ziZP · 2022-10-24

**Confidence:** 4
**Correctness:** 3
**Technical Novelty And Significance:** 3
**Empirical Novelty And Significance:** 3
**Recommendation:** 6

**Clarity, Quality, Novelty And Reproducibility:**

### Clarity
The paper is generally clear although I think it is extremely cluttered. It is clear that the text on page 5 is well below the margin. There is also very little space between sections in general. It is okay to play a bit with the formatting to try to fit the paper into the page limit but this paper is definitely overdoing it. This is really unfortunate because I think there is a lot of content that could have been left out or at least made more concise like the introduction and the background sections. Also, Section 5.3 doesn’t add much to the story of the paper and could be moved to the Appendix.

### Novelty
The idea of using exploration to improve generalization seems novel. The proposed exploration method is not very novel.

### Reproducibility
The code was not included in the submission which makes it hard to reproduce the results. The authors say that the code will be provided during the discussion.


**Strength And Weaknesses:**

### Strengths

I really like the idea of casting the generalization problem as an exploration problem. I agree that the standard RL formulation may be too task-centered, which makes it difficult for the agent to learn about the environment. The paper shows that this narrow focus may negatively affect generalization.

### Weaknesses

I am quite confused about what the true objective of this paper really is. The paper starts by claiming that any effective exploration strategy can improve generalization. Then, it proposes a method that targets exploration. I am not certain whether the proposed method is somehow addressing generalization directly (didn’t see any evidence for this in the paper) or if it is just another exploration method. If the latter, then the paper makes two separate contributions:

 (1) The hypothesis that exploration can improve generalization in CMDPs, which is interesting and seems to be partly supported by the experiments.

 (2) A method for encouraging exploration that does not seem very novel since it is based on existing ideas, and whose performance should be compared with other exploration methods.


### Questions/Suggestions

The abstract and introduction state that policy-based methods perform better than value-based methods in CMDPs. The authors identify exploration as the main reason for value-based methods not being able to generalize well in CMDPs. Is this suggesting that the reason policy-based methods outperform value-based methods is mainly that the former are more effective at exploring?

In order to be able to properly assess the novelty of the method, Section 4 should list only the ideas that are original and move those that belong to previous work to the background section. From what I understand, only equalized exploration is original here. I believe this idea is interesting and could be highlighted and further evaluated in the experiments.

Have you investigated the reasons why EDE performs better in terms of median/IQM but falls behind IDAAC when looking at the mean?


**Summary Of The Paper:**

This paper argues that effective exploration strategies can improve generalization in contextual MDPs (CMDPs). The paper proposes a method called exploration via distributional ensemble (EDE), which uses a deep ensemble to (1) disentangle epistemic uncertainty from aleatoric uncertainty and (2) encourage exploration by trying to reduce the epistemic uncertainty. EDE is shown to achieve SOTA for value-based methods in two popular benchmarks.

**Summary Of The Review:**

As mentioned before, this paper makes two separate contributions: (1) claiming that generalization can be improved with better exploration, and (2) proposing an exploration method. I don’t think it is fair to asses the paper as a whole since the contributions are somewhat orthogonal. I believe that the first contribution, although certainly interesting, is insufficient for acceptance in its current form. The second contribution does not seem very novel since the method is just a combination of previous solutions. Moreover, in order to determine its value, the method should be compared against other exploration methods.

My suggestion to the authors is to focus on one of the two. Either try to develop the first idea, providing further analysis and experiments (using different exploration strategies) to show that bad exploration is indeed the cause for poor generalization, or work on the exploration method and show that it can outperform other exploration strategies. This last direction is probably riskier since there are many works that focus on this problem.

---

> ### Author Response · Authors · 2022-11-09
> **Response to Reviewer ziZP (1/2)**
>
> We thank the reviewer for their thoughtful comments and useful suggestions for improving the paper. We were glad to hear they “really like the idea of casting the generalization problem as an exploration problem” and share our enthusiasm for this direction.
>
> ### Contribution
>
> >I am quite confused about what the true objective of this paper really is.
>
> To reiterate, our paper makes the following contributions: 1) identifies exploration as a key factor for generalization to new environments in contextual MDPs (CMDPs), 2) supports this hypothesis using a didactic example in a tabular CMDP whose aim is to motivate our work; 3) proposes an exploration approach that targets the agent’s epistemic uncertainty about the environment; 4) demonstrates that our method improves generalization and compares favorably with existing exploration methods as well as numerous ablations, being the first value-based method to achieve SOTA on Procgen and Crafter. We modified the introduction to clarify our contributions.
>
> 3 follows naturally from 2 because even though 2 suggests that effective exploration should help in the tabular setting, prior uncertainty-based exploration methods do not significantly help performance in CMDPs. The potential reason is that unobserved context induces significant amounts of aleatoric uncertainty which necessitate both better uncertainty quantification and disentanglement of different uncertainties, which is not a problem for singleton environments where generalization to new environments is not needed. **These two contributions are not independent as 2 is vacuous if we cannot show exploration helps generalization on real problems beyond the tabular setting.** 3 complements 2 by extending the intuition to challenging CMDPs that require representation learning because naively doing so does not work well. We’ve added discussion to this point at the beginning of Sec 4.
>
> >The paper starts by claiming that any effective strategy can improve generalization.
>
> We never claim that any exploration approach improves generalization which would of course be very incorrect. Rather, we claim that the exploration strategy employed affects the agent’s ability to generalize to new environments. We have now made this more explicit in the introduction.
>
> In fact, we compare our approach with 5 popular exploration methods for value-based algorithms, namely **Boostrapped DQN** (Osband & Van Roy, 2015), **UCB** (Chen et al. 2017), **ez-greedy** (Dabney et al. 2021), **UA-DQN** (Clements et al. 2019), and **NoisyNet** (Fortunato et al. 2017). Our results show that indeed the exploration strategy used by the agent plays an important role in how well it generalizes. Naively applying prior exploration methods without further modifications isn’t enough to improve generalization in this setting (see Figure 4). Section 5.1 contains a more detailed discussion on potential reasons for why prior exploration methods fall short in improving generalization.
>
> > Performance should be compared with other exploration methods.
>
> As mentioned above, we compare our approach with 5 popular exploration methods that have been shown to work well in singleton MDPs like Atari, as well as numerous ablations, resulting in a total of 12 value-based baselines (see Figure 4). We believe our study is extensive enough to demonstrate the superiority of our approach.
>
> >“My suggestion to the authors is to focus on one of the two [contributions].”
>
> Respectfully, we don’t believe that separating our contributions or only focusing on one of them would improve the paper. As mentioned above, the two contributions are complementary to each other. We provide evidence that better exploration leads to better generalization in a toy setting. In addition, we extensively compare our approach with other exploration methods and demonstrate its superiority, further supporting our hypothesis that how the agent explores its environment plays a key role in its ability to generalize to new environments. However, we are happy to discuss this point further if the reviewer has additional concerns or suggestions.

---

> > ### Author Response · Authors · 2022-11-09
> > **Response to Reviewer ziZP (2/2)**
> >
> > ### Novelty
> >
> > > The proposed exploration method is not very novel.
> >
> > We respectfully disagree with this assessment. As the reviewer also notes, the insight that exploration improves generalization on CMDPs is novel. Our paper corroborates this insight by showing that more effective exploration methods improve generalization to new environments on two different deep RL benchmarks. While individual parts of our proposed approach have been used in prior work, this particular combination is novel. In section, 5.1, we run an extensive ablation study and demonstrate the importance of each design choice and component of our algorithm. As shown in Figure 4, naively applying or combining prior exploration methods are not sufficient to significantly improve generalization in CMDPs.
> >
> > > Section 4 should list only the ideas that are original and move those that belong to previous work to the background section.
> >
> > We believe that the method section should be self-contained. The final paragraph of Section 4 summarizes our algorithm and highlights which parts are new and which parts are from prior work. We have added more details throughout Section 4 to further clarify which components of our algorithm are original by adding “we propose” to the original contribution where applicable. We hope that these changes have made our contributions clearer.
> >
> > ### Minor
> >
> > > The code was not included in the submission which makes it hard to reproduce the results.
> > >
> > Apologies for the delay, we wanted to clean up the code before uploading it. We have now added the code in the supplementary material, along with instructions for reproducing our results.
> >
> > >Very little space between sections
> >
> > Thank you for the suggestion — we added more space throughout the paper to make it less cluttered.
> >
> > > Section 5.3 doesn’t add much to the story
> >
> > Thank you for the suggestion — we moved Section 5.3 to Appendix E.
> >
> >
> > > I am not certain whether the proposed method is somehow addressing generalization directly (didn’t see any evidence for this in the paper) or if it is just another exploration method.
> >
> > Our method focuses on exploration and doesn’t aim to directly address generalization or representation learning in the conventional sense. As highlighted in the paper, most prior methods for improving generalization are based on representation learning (e.g. Song et al. 2020, Zhang et al. 2020, Cobbe et al. 2018, Laskin et al. 2020, Raileanu et al. 2020), while we explicitly focus on how exploration affects generalization. Exploration is a unique feature of the RL problem which doesn’t show up in classical deep supervised learning, and this approach is complementary to the supervised learning inspired approaches for improving generalization. We added a few sentences in the introduction to emphasize this point. We also hope this work would encourage future works to focus on ways of improving generalization in RL beyond representation learning.
> >
> > >Is this suggesting that the reason policy-based methods outperform value-based methods is mainly that the former are more effective at exploring?
> >
> > We do not claim that policy-gradient methods outperform value-based methods because they explore better but only claim that value-based methods can benefit from better exploration. We have edited the introduction to make this more explicit. We have also added a new set of experiments on Sarsa$(\lambda)$ in Sec 3 that suggests the reason why PG does better may be the whole trajectory update. This is somewhat orthogonal to exploration and PG may still benefit from better exploration, which is out of the scope of current work.
> >
> > > reasons why EDE performs better in terms of median/IQM but falls behind IDAAC when looking at the mean?
> > >
> > The median and IQM are more robust to outliers than the mean, so they are less affected by games where performance is very good or very bad. As shown in Appendix D, there are certain games where value-based approaches including EDE still fall behind policy optimization ones like IDAAC (although the reverse is also true but for a different set of games), which could explain the results. Agarwal et al. 2021 recommends the use of IQM and median rather than mean since they have lower uncertainty so they are more trustworthy for these types of experiments.
> >
> > ### Summary
> > We thank the reviewer again for their valuable feedback that has helped us further improve our paper. We hope our response and paper updates have adequately addressed your concerns and that you will consider increasing your support for our paper. If you have any outstanding concerns, please don’t hesitate to let us know so we can discuss them and understand what stands between us and a recommendation for acceptance.

---

> ### Author Response · Authors · 2022-11-15
> **Have the revision and response addressed the concerns?**
>
> Dear Reviewer ziZP,
>
> Thank you again for the review and questions. We hope our response and revision have clarified them and make the paper’s core message, contribution, and how the two parts of the paper relate to each other much clearer. Since the discussion period is closing at the end of the week, we were wondering if you have any further concerns or questions? If not, we would be happy to continue the discussion and/or revise the paper.

---

> > ### Comment · Reviewer_ziZP · 2022-12-05
> > **Response to authors**
> >
> > I want to thank the authors for addressing my concerns. After reading the other reviews and the authors' responses, I decided to raise my score to 6. The main reason for this is that, in my review, I incorrectly pointed out that the paper did not compare the method against other exploration baselines.
> >
> > That being said, I am still uncertain about what the objective and main contribution of this paper truly is (see points 1 and 2 in my review). In their response the authors say:
> >
> > > To reiterate, our paper makes the following contributions: 1) identifies exploration as a key factor for generalization to new environments in contextual MDPs (CMDPs), 2) supports this hypothesis using a didactic example in a tabular CMDP whose aim is to motivate our work; 3) proposes an exploration approach that targets the agent’s epistemic uncertainty about the environment; 4) demonstrates that our method improves generalization and compares favorably with existing exploration methods as well as numerous ablations, being the first value-based method to achieve SOTA on Procgen and Crafter. We modified the introduction to clarify our contributions.
> >
> > However, as stated in my original review, the proposed exploration method does not target generalization specifically. This was confirmed by the authors in their response. This raises the question of whether other exploration methods would also produce similar results. I am not an expert in exploration in RL but, given how many methods have been proposed over the years to target exploration, I highly doubt that the empirical results provided here are sufficient to conclude that the proposed method is the best (i.e. of all other exploration methods) at improving generalization. If this was not the case, the method could be replaced by some other, which makes the 3rd and 4th contributions listed above vacuous.
> >
> > I believe the real problem here is not that the paper is not sufficiently novel but that too much emphasis is given to the method. I think this distracts attention from the true contribution (i.e. showing that exploration improves generalization).

---

> > > ### Author Response · Authors · 2022-12-05
> > > **On the role of our proposed method**
> > >
> > > Thank you for getting back to us and raising the score! Indeed our main goal is to highlight the importance of exploration for generalization and we believe that our revision has made this clearer. We emphasize this point in the introduction and also later in the text. If you have any suggestions, we would be happy to incorporate them, but ultimately we believe it's important to describe the method we use and how it differs from prior methods so the paper is self-contained.
> > >
> > > Our proposed method targets exploration using epistemic uncertainty and our claim is that effective exploration helps generalization. As such, it targets generalization as the result. Further, we would like to bring the reviewer’s attention to [1] which was published in ICLR 2021 (so it’s recent). In this work, the authors found that **many exploration methods have overfit to environments like Montezuma’s revenge and do not do better than $\epsilon$-greedy on all the Atari games**. In fact, out of all the advanced exploration methods they tested, **only NoisyNet reliably outperforms $\epsilon$-greedy on Atari** (See Figure 4 of [1]). This is consistent with our observations that NoisyNet is the strongest baseline, and our method outperforms NoisyNet by a large margin on Procgen. We believe that this suggests that EDE should outperform other exploration methods. Unfortunately, we are unable to add new experiments to the paper now but if there is a particular method you want to see, we are more than happy to add the comparison in the revision.
> > >
> > > From a scientific perspective, EDE shows the importance of accurately separating epistemic and aleatoric uncertainties for generalization on CMDPs which to our knowledge has not been highlighted before. We believe this is important for future works in this direction.
> > >
> > > We hope this clarifies the role of the proposed method in the paper.
> > >
> > > **Reference**
> > >
> > > [1] On Bonus-Based Exploration Methods in the Arcade Learning Environment. Taiga et al. ICLR 2021.

---

### Author Response · Authors · 2022-11-09
**Common Response to All Reviewers**

We thank all the reviewers for their valuable feedback that has helped us further improve our paper. We were glad they found our paper to be “of importance to the RL generalization community”, “well-written”, with “comprehensive experiments” and “significant empirical improvements”.

We responded to the concerns of each reviewer below in separate replies and have updated the draft accordingly. We also added the code in the supplementary material.

Overall, it seems like the main concerns voiced by the reviewers are related to the paper’s (1) framing (reviewer ziZP, iatu, and Wm6P) and (2) novelty (reviewers ziZIP and R7TS).


**On Framing.** To reiterate, our paper makes the following contributions: 1) identifies exploration as a key factor for generalization to new environments in contextual MDPs (CMDPs), 2) supports this hypothesis using a didactic example in a tabular CMDP whose aim is to motivate our work; 3) proposes an exploration approach that enables efficient exploration in CMDPs that require function approximation via decomposing the agent’s epistemic and aleatoric uncertainty about the environment; 4) demonstrates that our method improves generalization and compares favorably with existing exploration methods as well as numerous ablations, being the first value-based method to achieve SOTA on Procgen and Crafter.

_On Exploration for Generalization._ Our goal is to improve the effectiveness of value-based methods (particularly those based on DQN) in CMDPs. We hypothesize that the reason for their poor performance in CMDPs, despite their strong performance in singleton MDPs, is partly due to insufficient exploration of the training environments. We empirically demonstrate that more sophisticated exploration methods based on uncertainty improve the generalization of value-based algorithms in both tabular CMDPs (used as a motivating example) and two challenging generalization benchmarks, Procgen and Crafter.

_On Policy-Optimization vs Value-Based_. While we mention that policy-optimization methods have been shown to be more effective than value-based methods in CMDPs, we **do not** claim this is due to better exploration. In fact, we suspect there are likely other reasons (we have added discussion of this point in Sec 3). Gaining a better understanding about this is an exciting future direction, but outside the scope of this paper. We have edited the abstract and introduction to better reflect the main focus of our paper, which is on improving the test performance of value-based methods in CMDPs by employing more effective exploration strategies, rather than comparing the exploration of policy-optimization and value-based algorithms.



_On Tabular Experiments_. The tabular experiments are meant as a motivational example and we do not claim it captures every aspect of more complex benchmarks such as Procgen. Indeed, it would be disingenuous to claim that tabular environments capture everything about deep RL. Nonetheless, we do believe they are helpful for building intuition. Our goal is to highlight that the problem of generalization in (deep) RL goes beyond representation learning (as typically treated in prior work) because we do not have to resort to function approximation in the tabular setting.

**On Novelty**. Our paper is first to highlight that the agent’s exploration strategy plays a key role in its ability to generalize to new environments when trained on CMDPs like Procgen. This contrasts with most prior methods which treat the problem as a representation learning problem. Our experiments in both tabular CMDPs as well as challenging deep RL benchmarks support this claim. While some individual parts of our proposed method have been used in various prior works, this particular combination and the conceptual connection of using them to improve generalization in CMDPs is novel and the main contribution of our work.

Finally, the specific technical innovations of EDE compared to prior methods are: 1) using deep ensembles for better estimation of different uncertainties (previous works use MAP sampling), 2) combining these uncertainties with UCB (previous works either use UCB with no uncertainty disentanglement, or use Thompson sampling with uncertainty disentanglement), and 3) proposing temporally extended exploration for diversity in data. In Section 5.1, we run an extensive ablation study and demonstrate the importance of each design choice and component of our algorithm. As shown in Figure 4, naively applying or combining existing exploration methods is not sufficient to significantly improve generalization on these benchmarks. That is, prior methods would not improve generalization on these benchmarks nearly as much as our method.

---

> ### Author Response · Authors · 2022-11-09
> **Common Response to All Reviewers Part 2**
>
> Following your feedback, we made the following updates to the paper (highlighted in blue) and uploaded the revised draft to OpenReview:
> 1. Edited the introduction and abstraction to better convey our core message.
> 2. Moved sec 5.3 to appendix E
> 3. Added more detailed descriptions of policy gradient in the background section.
> 4. Added Sarsa$(\lambda)$ with $\epsilon$-greedy to the tabular experiment (sec 3) to illustrate PG’s improvement may be attributed to full-trajectory update. Sarsa$(\lambda)$ still underperforms UCB and benefit from exploration.
> 5. Added discussion of “better training performance -> better test performance” to sec 3. This was originally in 5.3.
> 6. Added an example of how the scenario in tabular applies to Procgen through a concrete example based on the game bigfish in Appendix I.
> 7. Added more explicit transition between sec 3 and 4 to highlight the connection between the two.
> 8. Added more details for uncertainty estimation in sec 4 to make it more self-contained.

---

### Decision · Program_Chairs · 2023-01-20

**Decision:**

Reject

**Justification For Why Not Higher Score:**

See the main concern described above.

**Justification For Why Not Lower Score:**

N/A.

**Metareview: Summary, Strengths And Weaknesses:**

The main contribution of this work lies in proposing an exploration strategy called exploration via distributional ensemble for improving generalization in contextual MDPs.

After reviewing the authors' response and an active discussion, the reviewers have agreed that the combination of ideas is sufficiently novel.

The authors have initially given some reviewers the impression that policy-based methods perform better than value-based methods in CMDPs. They have since rectified this claim and explained what they mean.

However, a main concern of the reviewers remains, that is, the connection between "more effective exploration during training" and "better generalization in testing" is still not established well: For example, the illustrative bigfish example seems to be a hypothesis that is not empirically validated/supported sufficiently well and hence questionable. The authors are strongly encouraged to revise their work to substantiate this claim in a compelling way.

**Summary Of Ac-Reviewer Meeting:**

There is sufficient **written** discussion generated on the OpenReview discussion forum to the extent of being able to reach a consensus on the recommendation. Hence, there is no need for a meeting.